# Some New Bounds for the Inverse Sum Indeg Energy of Graphs

**Fengwei Li** [1,2,*]**, Qingfang Ye** [1] **and Hajo Broersma** [2]

[1] College of Basic Science, Ningbo University of Finance & Economics, Ningbo 315175, China; fqy-y@163.com
[2] Faculty of EEMCS, University of Twente, P.O. Box 217, 7500 AE Enschede, The Netherlands; h.j.broersma@utwente.nl
[*] Correspondence: fengwei.li@hotmail.com

**Abstract:** Let $G$ be a (molecular) graph with $n$ vertices, and $d_i$ be the degree of its $i$-th vertex. Then, the inverse sum indeg matrix of $G$ is the $n \times n$ matrix $\mathcal{C}(G)$ with entries $c_{ij} = \frac{d_i d_j}{d_i + d_j}$, if the $i$-th and the $j$-th vertices are adjacent and 0 otherwise. Let $\mu_1 \geq \mu_2 \geq \ldots \geq \mu_n$ be the eigenvalues of $\mathcal{C}$ arranged in order. The inverse sum indeg energy of $G$, $\varepsilon_{isi}(G)$ can be represented as $\sum_{j=1}^{n} |\mu_i|$. In this paper, we establish several novel upper and lower sharp bounds on $\mu_1$ and $\varepsilon_{isi}(G)$ via some other graph parameters, and describe the structures of the extremal graphs.

**Keywords:** inverse sum indeg index; energy; ISI matrix; ISI energy





## 1. Introduction

In the whole article, let $G$ be an undirected simple finite graph with the collection of vertex $V(G) = \{u_1, u_2, \ldots, u_n\}$ and edge $E(G)$. We use $u_i \sim u_j$ to represent two adjacent vertices $u_i$ and $u_j$, and $u_i u_j$ to indicate the edge in $E(G)$ with two end vertices $u_i$ and $u_j$. The degree of vertex $u_i$ is represented by $d_i$, and a *j-vertex* denotes a vertex of degree $j$. An *n-vertex graph* denotes the graph of order $n$. We call $\delta = min_{1 \leq i \leq n}\{d_i\}$ and $\Delta = max_{1 \leq i \leq n}\{d_i\}$ the *minimum degree* and the *maximum degree* of an $n$-vertex graph $G$, respectively. Furthermore, $(n, m)$-*graph* denotes the graph of order $n$ and size $m$. Analogously, let $(n, m, \delta, \Delta)$-*graph* express the graph of order $n$, size $m$, minimum degree $\delta$ and maximum degree $\Delta$. Let $tr(M)$ stand for the *trace* of matrix $M$. An *independent set* of $G$ is a subset $S \subset V(G)$, so that in the induced subgraph $G[S]$ exist no edges. Furthermore, $\alpha(G)$ signifies the *independence number* of $G$ [1].

A graph possessing only $r$-vertex is named as an $r$-*regular* graph. Let $s \neq t$ be two positive integers; a graph $G$ is called an $(s, t)$-*semiregular* if it possesses either $s$-vertex or $t$-vertex, and there exists no fewer than one $s$-vertex and one $t$-vertex.

Let $A = A(G)$ stand for the adjacency matrix of graph $G$. We call set $SpA(G) = \{\lambda_1, \lambda_2, \ldots, \lambda_n\}$ the $A$-*spectrum* of $G$. We list the eigenvalues of $A(G)$ in order $\lambda_1 \geq \lambda_2 \geq \ldots \geq \lambda_n$, and call the maximum eigenvalue, $\lambda_1$, the *spectral radius* of $G$.

Among the various utilizations of graph theory in chemistry, the close relationship between the graph eigenvalues and the molecular orbital energy levels of $\pi$ electrons in conjugated hydrocarbons is the most significant. In theoretical chemistry, with the help of the Hückel theory, the $\pi$-electron energy of conjugated carbon molecules is found to be consistent with the energy [2–4]. Accordingly, graph energy has rich meanings, both in theory and practice.

The *energy* [3,5,6] of the graph $G$ is defined as

$$\varepsilon = \varepsilon(G) = \sum_{i=1}^{n} |\lambda_i| . \tag{1}$$

This concept was introduced by Gutman [5] and is frequently studied in chemistry. The energy of chemically relevant molecular graphs was shown to be quantitatively related

with the experimentally determined heats of formation and other measures of the thermo-dynamic stability of underlying conjugated compounds. Following in-depth research, it was found that this graph parameter can be successfully utilized in many fields, not only in chemistry [3,7,8]. In consideration of the successful development of the mathematical theory of graph energy, many extended graph energies have been gradually proposed based on the eigenvalues of other graph matrices, such as the (first) Zagreb matrix [9], the harmonic matrix [10], etc.; see [3,9,11–17] and some more recent results to be found in [18–20].

In [21], Gutman et al. introduced the *Randić matrix*, $R(G) = (r_{ij})_{n \times n}$, of a graph $G$, where $r_{ij} = \frac{1}{\sqrt{d_i d_j}}$ if $v_i v_j \in E(G)$ and is 0 otherwise. Denote its eigenvalues by $\rho_1 \geq \rho_2 \geq \ldots \geq \rho_n$. Then, in analogy to Equation (1), the *Randić energy* is defined as

$$RE = RE(G) = \sum_{i=1}^{n} |\rho_i| . \tag{2}$$

The *extended adjacency matrix* of graph $G$, denoted by $A_{ex} = A_{ex}(G)$, was put forward by Yang et al. [22] and is defined so that its $(i, j)$ entry is equal to $\frac{1}{2}(\frac{d_i}{d_j} + \frac{d_j}{d_i})$ if $v_i v_j \in E(G)$ and is 0 otherwise. The *extended graph energy* is defined as

$$\varepsilon_{ex} = \varepsilon_{ex}(G) = \sum_{i=1}^{n} |\zeta_i| . \tag{3}$$

where $\zeta_1 \geq \zeta_2 \geq \ldots \geq \zeta_n$ are the ordered eigenvalues of $A_{ex}$.

In [12], Das et al. gave lower and upper bounds on the extended spectral radius $\zeta_1$ and the extended energy $\varepsilon_{ex}$ of graphs and the respective extremal graphs were characterized.

Topological indices are of great importance to mathematical chemistry. A great deal of topological indices, such as the Randić index [23], atom–bond-connectivity index [24], sum-connectivity index [25], augmented Zagreb index [26], the eccentric-connectivity index [27], Zagreb indices [6,28], the general eccentric-connectivity index [29], the general degree-eccentricity index [30], etc., were introduced to reveal the properties of organic compounds from different aspects. One of those numerical descriptors, the *inverse sum indeg index* (*ISI index* for short) is an especially interesting vertex-degree-based topological index, which is defined as

$$ISI(G) = \sum_{v_i v_j \in E(G)} \frac{d_i d_j}{d_i + d_j}.$$

In 2010, Vukičević and Gašperov [31] proposed the ISI index, which can distinctively forecast the overall surface area of octane isomers.

Similar to the Randić matrix and the extended adjacency matrix, Li et al. [32] and Zangi et al. [33] defined the *inverse sum indeg matrix* (*ISI matrix* for short) $\mathcal{C} = \mathcal{C}(G)$ of a graph $G$ as the matrix with entries:

$$c_{ij} := \begin{cases} \frac{d_i d_j}{d_i + d_j}, & if \ v_i v_j \in E(G) \\ 0, & otherwise, \end{cases}$$

respectively. Note that $\mathcal{C}$ is a modification of the classical adjacency matrix involving the degrees of the vertices.

Denote by $\mu_1 \geq \mu_2 \geq \ldots \geq \mu_n$ the ordered eigenvalues of $\mathcal{C}$. The multiset $Sp_{isi} = Sp_{isi}(G) = \{\mu_1, \mu_2, \ldots, \mu_n\}$ will be called the *ISI spectrum* of the graph $G$. We say $\mu_1$ is the *ISI spectral radius* of $G$. Extending the energy concept to the ISI matrix, the *ISI energy* of a graph $G$ can be defined as follows

$$\varepsilon_{isi} = \varepsilon_{isi}(G) = \sum_{i=1}^{n} |\mu_i| . \tag{4}$$

In recent years, researchers have found that graph energy and its variants have diverse, amazing and, to some extent, unanticipated utilizations in crystallography [34,35], the analysis and comparison of protein sequences [36,37], the theory of macromolecules [38,39], network analysis [40–45], and so on. It is noted that there is a very close relationship between the $\varepsilon(G)$ and $\varepsilon_{isi}(G)$ of graphs. Therefore, we can use $\varepsilon_{isi}(G)$ to obtain the $\varepsilon(G)$ of numerous kinds of graphs. Consequently, it has not only theoretical importance, but also practical significance in $\varepsilon_{isi}(G)$ research.

In 2018, Das et al. [13] normalized almost all kinds of degree-based graph energies into a unified form, and they derived some bounds on these energies of graphs. In this paper, novel bounds for $\mu_1$ and $\varepsilon_{isi}(G)$ were acquired, and these bounds can not be deduced from the results in [13].

In this paper, we also need the *general Randić index*

$$R_{\frac{1}{2}}(G) = \sum_{v_i v_j \in E(G)} \sqrt{d_i d_j} \, ,$$

which was introduced by Bollobás and Erdős [46], and the *Zagreb indices* introduced by Gutman and Trinajstic [6] in 1972. The first and second Zagreb indices of a graph $G$ are denoted by $M_1(G)$ and $M_2(G)$, respectively, and defined as

$$M_1(G) = \sum_{v_i \in V(G)} d_i^2, M_2(G) = \sum_{v_i v_j \in E(G)} d_i d_j \, .$$

We structure this paper in four parts. Some subsequently used definitions, notations and results are offered in Section 2. Section 3 gives some bounds for $\mu_1$ and characterizes the corresponding graphs. Several novel bounds on $\varepsilon_{isi}(G)$ are established in Section 4.

## 2. Preliminaries

In this part, we give some lemmas which will come in handy in later parts.

**Lemma 1** ([32]). *For any connected graph G and every edge $v_i v_j \in E(G)$, we have*

$$\frac{\delta}{2} \leq \frac{d_i d_j}{d_i + d_j} \leq \frac{\Delta}{2} \, . \tag{5}$$

*the equality in left and right hands are both attained iff G is regular.*

**Lemma 2** ((Cauchy–Schwarz inequality) [47]). *Let W and Z be two n-dimension vectors with elements $w_i \in \mathbb{R}$ and $z_i \in \mathbb{R}(1 \leq i \leq n)$, respectively. Then*

$$\left( \sum_{i=1}^{n} w_i z_i \right)^2 \leq \sum_{i=1}^{n} w_i^2 \sum_{i=1}^{n} z_i^2 \, , \tag{6}$$

*with equality iff there is a real number d satisfying that $w_j = d z_j$ $(1 \leq j \leq n)$.*

**Lemma 3** ((Chebyshev's inequality) [48]). *For two sequences of $w_i \in \mathbb{R}$ and $z_i \in \mathbb{R}(1 \leq i \leq n)$, such that $w_1 \leq w_2 \leq \ldots \leq w_n$ and $z_1 \leq z_2 \leq \ldots \leq z_n$, we have*

$$\left( \sum_{i=1}^{n} w_i \right) \left( \sum_{i=1}^{n} z_i \right) \leq n \sum_{i=1}^{n} w_i z_i \, , \tag{7}$$

*the equality is obtained iff $w_1 = w_2 = \ldots = w_n$ or $z_1 = z_2 = \ldots = z_n$.*

**Lemma 4** ([49]). *Let $w_i (1 \leq i \leq n)$ be real numbers satisfying $w_n \leq w_{n-1} \leq \ldots \leq w_1$. Then*

$$\frac{\sum_{i=1}^n w_i}{n} + \sqrt{\frac{1}{n(n-1)} \sum_{i=1}^n \left( w_i - \frac{\sum_{i=1}^n w_i}{n} \right)^2} \leq w_1. \tag{8}$$

**Lemma 5** ([50]). *Let $G$ be a connected $(n, m, \delta, \Delta)$-graph, for any $v_i v_j \in E(G)$, we have*

$$\frac{2\sqrt{\delta\Delta}}{\Delta + \delta} \leq \frac{2\sqrt{d_i d_j}}{d_i + d_j} \leq 1.$$

*The equality in the left hand is attained iff $G$ is $(\Delta, \delta)$-semiregular or regular; the equality in the right hand is achieved iff $G$ is regular.*

Recall that for any $n$-order square matrices $M = (s_{jk})$ and $N = (t_{jk})$, if $s_{jk} \geq t_{jk}$ holds for any $j, k$, then $M \geq N$.

**Lemma 6** ([51]). *If $M \geq N$ for any two symmetric, non-negative $n$-order square matrices $M$ and $N$, then $\rho_1(M) \geq \rho_1(N)$, where $\rho_1$ is the maximum eigenvalue.*

**Lemma 7** ((Interlacing Lemma) [52]). *If $M$ is a symmetric $n$-order square matrix, and $M_j$ is the $j \times j$ submatrix of $M$, then, for any integer $k$, $1 \leq k \leq j$,*

$$\rho_{n-k+j}(M) \leq \rho_i(M_j) \leq \rho_k(M), \tag{9}$$

*where $\rho_k(M_j)$, $\rho_k(M)$ are the $k$-th greatest eigenvalue of $M$ and $M_j$, respectively.*

**Lemma 8** ([53]). *If $G$ is an $n$-vertex graph having degree collection $d_i (1 \leq j \leq n)$, then*

$$\lambda_1 \geq \sqrt{\frac{\sum_{i=1}^n d_i^2}{n}}, \tag{10}$$

*the equality is achieved iff $G$ is regular or semiregular.*

**Lemma 9** ((Rayleigh–Ritz) [54]). *Let $M$ be a real symmetric $n$-order square matrix having eigenvalues $\rho_1 \geq \rho_2 \geq \ldots \geq \rho_n$, then, for a nonzero vector $\mathbf{y}$,*

$$\rho_1 \geq \frac{\mathbf{y}^T M \mathbf{y}}{\mathbf{y}^T \mathbf{y}}, \tag{11}$$

*the equality holds iff $\mathbf{y}$ is an eigenvector for $\rho_1$ of $M$.*

**Lemma 10** ([55]). *For any $n$-vertex graph $G$, $\varepsilon_{isi}(G) = 0$ iff $G \cong \overline{K_n}$.*

**Lemma 11** ([56]). *For any $(n, m, \delta, \Delta)$-graph $G$ such that $\delta \geq 1$,*

$$\lambda_1 \leq \sqrt{2m - \delta(n-1) + (\delta - 1)\Delta}, \tag{12}$$

*the equality is acquired iff $G$ is regular, or the union of $K_{1,n-3}$ and $K_2$, or the union of a regular graph possessing smaller vertex degree and a complete graph.*

**Lemma 12.** *Let $G$ be an $n$-vertex connected graph. Then $\mu_1 > \mu_2$.*

**Proof.** Suppose the result is false. Let $\mathbf{u}$ and $\mathbf{v}$ be eigenvectors corresponding to $\mu_1$ and $\mu_2$, respectively. Note that the fact that $G$ is connected. If $\mu_1 = \mu_2$, by Perron–Frobenius

theorem, all elements of **u** are positive. Since $\mu_1 = \mu_2$, any linear combination of **u** and **v** has to be an eigenvector for $\mu_1$. In this way, one element of the vector can be adjusted to zero easily, a contradiction. $\square$

**Lemma 13.** *Let G be an n-vertex graph.Then,* $|\mu_1| = |\mu_2| = \ldots = |\mu_n|$ *if and only if* $G \cong \overline{K_n}$ *or* $G \cong \frac{n}{2}K_2$.

**Proof.** For convenience, we use $I$ to stand for the collection of isolated vertices of $G$. First, we assume $|\mu_1| = |\mu_2| = \ldots = |\mu_n|$. Let $k = |I|$. If $k \geq 1$, then $\mu_1 = \mu_2 = \ldots = \mu_n = 0$, i.e., $G \cong \overline{K_n}$. Otherwise, $k = 0$. If $\Delta = 1$, then $d_i = 1 (1 \leq i \leq n)$ and therefore $G \cong \frac{n}{2}K_2$. Otherwise, $\Delta \geq 2$, then $G$ includes a connected component $H$ such that $|V(H)| \geq 3$. If $H \cong K_p$ $(p \geq 3)$, then $\mu_1(H) = \frac{(p-1)^2}{2} > \frac{p-1}{2} = \mu_2(H)$, a contradiction. Otherwise, $H \not\cong K_p$ $(p \geq 3)$, then $diam(H) \geq 2$. We suppose that $G$ contain an induced shortest path $P_m$, $m \geq 2$. Let $B$ be the principal submatrix of $\mathcal{C}$ indexed by the vertices of $P_m$ and then by the interlacing theorem 7 we obtain $\mu_2(H) \geq \mu_2(B) \geq 0$. Moreover, by Lemma 12, we know that $\mu_1(H) > \mu_2(H)$, a contradiction.

On the contrary, when $G \cong \overline{K_n}$ or $G \cong \frac{n}{2}K_2$, we have $|\mu_1| = |\mu_2| = \ldots = |\mu_n|$. $\square$

**Lemma 14** ([32]). *For any n-vertex graph G having vertices* $v_j (1 \leq j \leq n)$, *we have*

*(1)* $tr(\mathcal{C}) = 0$;

*(2)*

$$tr(\mathcal{C}^2) = 2 \sum_{v_i v_j \in E(G)} \frac{d_i^2 d_j^2}{(d_i + d_j)^2} ;$$

*(3)*

$$tr(\mathcal{C}^4) = \sum_{v_i \in V(G)} \left( \sum_{v_j \sim v_i} \frac{d_i^2 d_j^2}{(d_i + d_j)^2} \right)^2$$

$$+ \sum_{\substack{v_i, v_j \in V(G) \\ v_i \neq v_j}} d_i^2 d_j^2 \left( \sum_{\substack{v_k \in V(G) \\ v_k \sim v_i, v_k \sim v_j}} \frac{d_k^2}{(d_i + d_k)(d_k + d_j)} \right)^2 .$$

**Lemma 15** ( [9]). *For integers* $z_i \geq 0 (1 \leq i \leq n)$ *and* $s \geq 2$, *we get*

$$\sum_{i=1}^{n} (z_i)^s \leq \left( \sum_{i=1}^{n} z_i^2 \right)^{\frac{s}{2}} , \tag{13}$$

*the equality is obtained iff* $z_1 = z_2 = \ldots = z_n$.

**Lemma 16** ([57]). *Let G be an n-vertex connected graph. Then there exists just one positive eigenvalue in* $SpA(G)$ *iff G is isomorphic to* $K_{l_1, l_2, \ldots, l_s}$, $n = l_1 + l_2 + \ldots + l_s$.

**Lemma 17** ([9]). *Let* $y_i > 0 (1 \leq i \leq p)$ *in* $\mathbb{R}$. *Then*

$$\frac{p}{\frac{1}{y_1} + \frac{1}{y_2} + \ldots + \frac{1}{y_p}} \leq \sqrt[p]{y_1 y_2 \ldots y_p} . \tag{14}$$

The *nullity* $n_0(G)$ of a graph $G$ is the multiplicity of eigenvalue 0 in its adjacency spectrum.

**Lemma 18** ([48]). *For any* $n(n \geq 2)$*-vertex graph G,* $n_0(G) = n - 2$ *iff* $G \cong K_{s,t} \cup (n - s - t)K_1$, $s + t \leq n$.

**Lemma 19** ([55]). *For any graph G with components $G_i$, $(1 \le i \le k)$, we have*

$$\varepsilon_{isi}(G) = \sum_{i=1}^{k} \varepsilon_{isi}(G_i) .$$

### 3. Bounds for the ISI Spectral Radius

In this part, we establish several bounds for the ISI spectral radius $\mu_1$ of graphs.

We first present an upper bound on $\mu_1$ in terms of the maximum degree $\Delta$, minimum degree $\delta$, order $n$ and size $m$.

**Theorem 1.** *If G is an $(n, m, \delta, \Delta)$-graph so that $\Delta \ge \delta \ge 1$, then*

$$\mu_1 \le \frac{\Delta}{2}\sqrt{2m - \delta(n-1) + (\delta - 1)\Delta} \ , \tag{15}$$

*the equality is acquired iff G is a regular graph.*

**Proof.** Lemma 1 deduce that $\mathcal{C} \le \frac{\Delta}{2}A(G)$. Furthermore, by Lemmas 6 and 11, we have

$$\mu_1 \le \frac{\Delta}{2}\lambda_1 \le \frac{\Delta}{2}\sqrt{\Delta(\delta - 1) + 2m - (n-1)\delta}.$$

If $\mu_1 = \frac{\Delta}{2}\sqrt{\Delta(\delta - 1) + 2m - (n-1)\delta}$, from $\mathcal{C} = \frac{\Delta}{2}A(G)$, we have $\frac{d_i d_j}{d_i + d_j} = \frac{\Delta}{2}$ i.e., $d_1 = d_2 = \ldots = d_n = \Delta$. Therefore, G must be regular.

On the contrary, it is obvious that the equality in (15) holds when G is regular.
This completes the proof.　□

**Theorem 2.** *For any $(n, \delta)$-graph G,*

$$\mu_1 \ge \frac{\delta}{2}\sqrt{\frac{M_1}{n}} \ , \tag{16}$$

*the equality is acquired iff G is regular.*

**Proof.** From Lemma 1, we know that $\frac{d_i d_j}{d_i + d_j} \ge \frac{\delta}{2}$. Then, $\mathcal{C} \ge \frac{\delta}{2}A$. Furthermore, by Lemmas 6 and 8,

$$\mu_1 \ge \frac{\delta}{2}\lambda_1 \ge \frac{\delta}{2}\sqrt{\frac{\sum_{i=1}^{n} d_i^2}{n}} = \frac{\delta}{2}\sqrt{\frac{M_1}{n}} \ .$$

Now, let's assume that the equation holds in (16). Then aforementioned inequalities must be equalities. From $\mathcal{C} = \frac{\delta}{2}A$, we have $\frac{d_i d_j}{d_i + d_j} = \frac{\delta}{2}$ i.e., $d_1 = d_2 = \ldots = d_n = \delta$. Thus, G must be regular.

On the contrary, when G is regular, it can be easily proved that $\mu_1 = \frac{\delta}{2}\sqrt{\frac{M_1}{n}}$.　□

**Theorem 3.** *For any $(n, \Delta)$-graph G,*

$$\mu_1 \ge \frac{M_2}{n\Delta}, \tag{17}$$

*the equality is acquired iff G is regular.*

**Proof.** Let $\mathbf{x} = (x_1, x_2, \ldots, x_n)^T$ be a unit vector, $x_i \in \mathbb{R}$ $(1 \le i \le n)$. Then,

$$\mathbf{x}^T \mathcal{C} \mathbf{x} = 2 \sum_{v_i v_j \in E(G)} \frac{d_i d_j}{d_i + d_j} x_i x_j \ge 2 \sum_{v_i v_j \in E(G)} \frac{d_i d_j}{2\Delta} x_i x_j . \tag{18}$$

Set $\mathbf{x} = (\frac{1}{\sqrt{n}}, \frac{1}{\sqrt{n}}, \ldots, \frac{1}{\sqrt{n}})^T$. Then, from Lemma 9, we have

$$\mu_1 \geq \mathbf{x}^T \mathcal{C} \mathbf{x} \geq \frac{\sum_{v_i v_j \in E(G)} d_i d_j}{n \Delta} = \frac{M_2}{n \Delta}.$$

If $\mu_1 = \frac{M_2}{n \Delta}$, from (17), we have $d_1 = d_2 = \ldots = d_n = \Delta$. Furthermore, from $\mu_1 = \mathbf{x}^T \mathcal{C} \mathbf{x}$, we have that the vector $x = (\frac{1}{\sqrt{n}}, \frac{1}{\sqrt{n}}, \ldots, \frac{1}{\sqrt{n}})^T$ is an eigenvector for $\mu_1$. Thus, $G$ is certainly to be a regular graph.

On the contrary, the equality in (17) is achieved if $G$ is regular. $\quad\square$

Resembling the method in Theorem 3, an upper bound of $\mu_1$ can be gained on the basis of $\Delta$, $\delta$ and $R_{\frac{1}{2}}(G)$.

**Theorem 4.** *If $G$ is an $(n, m, \delta, \Delta)$-graph, we obtain*

$$\mu_1 \geq \frac{2\sqrt{\delta \Delta}}{n(\Delta + \delta)} R_{\frac{1}{2}}, \tag{19}$$

*the equality is acquired iff $G$ is certainly to be a regular graph.*

**Proof.** Assume that $\mathbf{x} = (x_1, x_2, \ldots, x_n)^T$ is a unit vector, $x_i \in \mathbb{R}$, $1 \leq i \leq n$.

$$\mathbf{x}^T \mathcal{C} \mathbf{x} = 2 \sum_{v_i v_j \in E(G)} \frac{d_i d_j}{d_i + d_j} x_i x_j$$

$$= 2 \sum_{v_i v_j \in E(G)} \frac{2\sqrt{d_i d_j}}{d_i + d_j} \frac{\sqrt{d_i d_j}}{2} x_i x_j$$

$$\geq \frac{2\sqrt{\delta \Delta}}{\Delta + \delta} \sum_{v_i v_j \in E(G)} \sqrt{d_i d_j} x_i x_j. \tag{20}$$

Set $\mathbf{x} = (\frac{1}{\sqrt{n}}, \frac{1}{\sqrt{n}}, \ldots, \frac{1}{\sqrt{n}})^T$. Then, from Lemma 9, we deduce

$$\mu_1 \geq \mathbf{x}^T \mathcal{C} \mathbf{x} \geq \frac{2\sqrt{\delta \Delta}}{n(\Delta + \delta)} \sum_{v_i v_j \in E(G)} \sqrt{d_i d_j} = \frac{2\sqrt{\delta \Delta}}{n(\Delta + \delta)} R_{\frac{1}{2}}. \tag{21}$$

If the equality holds in (19), we take it for granted that all above-mentioned inequalities must be equalities. We are aware of that $G$ is a $(\Delta, \delta)$-biregular or regular graph by (19). Furthermore, $\mu_1 = \mathbf{x}^T \mathcal{C} x$ deduce that vector $x = (\frac{1}{\sqrt{n}}, \frac{1}{\sqrt{n}}, \ldots, \frac{1}{\sqrt{n}})^T$ must be an eigenvector for $\mu_1$. Hence, $G$ is certainly to be a regular graph.

Conversely, it is easily inspected that $\mu_1 = \frac{2\sqrt{\delta \Delta}}{n(\Delta + \delta)} R_{\frac{1}{2}}$ under the condition that $G$ is a regular. $\quad\square$

We note that $R_{\frac{1}{2}}(G) = \sum_{v_i v_j \in E(G)} \sqrt{d_i d_j} \geq m\delta$. So, the following corollary can be easily got.

**Corollary 1.** *Let $G$ be an $(n, m, \delta, \Delta)$-graph. Then*

$$\mu_1 \geq \frac{2\delta m \sqrt{\Delta \delta}}{n(\delta + \Delta)},$$

*the equality is acquired iff $G$ is a regular graph.*

## 4. Bounds for the ISI Energy of Graphs

For convenience, we let $\gamma_1 \geq \gamma_2 \geq \ldots \geq \gamma_n$ as the absolute values of eigenvalues $\mu_i, 1 \leq i \leq n$, which are arranged in decreasing order. It is obvious that

$$\varepsilon_{isi}(G) = \sum_{i=1}^{n} |\mu_i| = \sum_{i=1}^{n} \gamma_i$$

and

$$tr(\mathcal{C}^2) = \sum_{i=1}^{n} \mu_i^2 = \sum_{i=1}^{n} \gamma_i^2 \ .$$

**Theorem 5.** *Let G be a graph of order n and size m, with minimum degree $\delta$ and maximum degree $\Delta$. Then*

$$\varepsilon_{isi}(G) \leq \frac{\Delta}{2}\sqrt{2mn} \ . \tag{22}$$

*The equality holds if and only if $G \cong \overline{K_n}$ or $G \cong \frac{n}{2}K_2$ .*

**Proof.** Bear in mind that $\gamma_1^2, \gamma_2^2, \ldots, \gamma_n^2$ form the eigenvalues of $\mathcal{C}^2$. Combined this fact with Lemma 2 we obtain

$$\varepsilon_{isi}(G) = \sum_{i=1}^{n} \gamma_i \leq \sqrt{\sum_{i=1}^{n} 1}\sqrt{\sum_{i=1}^{n} \gamma_i^2} = \sqrt{n tr(\mathcal{C}^2)} \ . \tag{23}$$

Lemma 13 implies that

$$tr(\mathcal{C}^2) = 2 \sum_{v_i v_j \in E(G)} \frac{(d_i d_j)^2}{(d_i + d_j)^2} \leq \frac{m\Delta^2}{2}.$$

Hence, we have

$$\varepsilon_{isi}(G) \leq \frac{\Delta}{2}\sqrt{2mn} \ .$$

The equality in (23) is clearly attained when $G \cong \overline{K_n}$ or $G \cong \frac{n}{2}K_2$.

If the equality in (22) is achived, then equality must hold in (23). So we have that $\gamma_1 = \gamma_2 = \ldots = \gamma_n$. Hence, $G \cong \frac{n}{2}K_2$ or $G \cong \overline{K_n}$. □

**Theorem 6.** *Let G be an n-vertex graph, we have*

$$\varepsilon_{isi}(G) \geq (n-1)\gamma_1 + \sqrt{tr(\mathcal{C}^2) - (n-1)\gamma_1^2} \ , \tag{24}$$

*and the equality is obtained iff $G \cong \overline{K_n}$ or $G \cong \frac{n}{2}K_2$ .*

**Proof.** Setting $w_i = \gamma_i, (i = 1, 2, \ldots, n)$, inequality (8) becomes

$$\frac{1}{n}\sum_{i=1}^{n} \gamma_i + \sqrt{\frac{1}{n(n-1)}\sum_{i=1}^{n}\left(\gamma_i - \frac{1}{n}\sum_{i=1}^{n} \gamma_i\right)^2}$$

$$= \frac{\varepsilon_{isi}(G)}{n} + \sqrt{\frac{1}{n(n-1)}\sum_{i=1}^{n}\left(\gamma_i - \frac{\varepsilon_{isi}(G)}{n}\right)^2}$$

$$= \frac{\varepsilon_{isi}(G)}{n} + \sqrt{\frac{1}{n(n-1)}\left(\sum_{i=1}^{n} \gamma_i^2 - \frac{2\varepsilon_{isi}(G)}{n}\sum_{i=1}^{n} \gamma_i + \sum_{i=1}^{n}\frac{(\varepsilon_{isi}(G))^2}{n^2}\right)}$$

$$= \frac{\varepsilon_{isi}(G)}{n} + \sqrt{\frac{1}{n(n-1)}\left(tr(\mathcal{C}^2) - \frac{(\varepsilon_{isi}(G))^2}{n}\right)} \leq \gamma_1.$$

Inequality (23) implies that $tr(\mathcal{C}^2) \geq \frac{(\varepsilon_{isi}(G))^2}{n}$. Hence,

$$(\varepsilon_{isi}(G))^2 - 2(n-1)\gamma_1\varepsilon_{isi}(G) \geq tr(\mathcal{C}^2) - n(n-1)\gamma_1^2 .$$

Assume that equality in (24) is achieved. Then $\gamma_i^2 = \frac{1}{n}\sum_{i=1}^n \gamma_i^2$ for any $1 \leq i \leq n$. So, we have $\gamma_1 = \gamma_2 = \ldots = \gamma_n$. By Lemma 13, we know that $G \cong \overline{K_n}$ or $G \cong \frac{n}{2}K_2$.

Conversely, the equality in (24) holds obviously for $G \cong \frac{n}{2}K_2$ or $G \cong \overline{K_n}$.

This completes the proof.　□

As a generalisation of the Shisha–Mond inequality, the Klamkin–McLenaghan inequality can be stated as follows.

**Lemma 20** ((Klamkin–McLenaghan inequality) [58]). *Let* $X = (x_1, x_2, \ldots, x_n)$ *and* $Y = (y_1, y_2, \ldots, y_n)$ *be n-tuples of non-negative real numbers satisfying* $0 \leq m \leq \frac{x_i}{y_i} \leq M$ *for each* $i \in \{1, 2, \ldots, n\}$, *and* $w_i \geq 0$. *Then,*

$$\frac{\sum_{i=1}^n w_i x_i^2}{\sum_{i=1}^n w_i x_i y_i} - \frac{\sum_{i=1}^n w_i x_i y_i}{\sum_{i=1}^n w_i y_i^2} \leq (\sqrt{M} - \sqrt{m})^2 . \tag{25}$$

**Theorem 7.** *Let G be a graph of order n. Then, the following inequality is valid:*

$$\varepsilon_{isi}(G) \geq \frac{1}{2}\left(\sqrt{4ntr(\mathcal{C}^2) + n^2(\sqrt{\gamma_1} - \sqrt{\gamma_n})^2} - n(\sqrt{\gamma_1} - \sqrt{\gamma_n})^2\right) . \tag{26}$$

*Equality is attained if and only if* $G \cong \overline{K_n}$ *or* $G \cong \frac{n}{2}K_2$.

**Proof.** Setting $x_i = \gamma_i$, $y_i = 1$, and $w_i = 1$, $(i = 1, 2, \ldots, n)$, then $0 \leq \gamma_n \leq \frac{x_i}{y_i} \leq \gamma_1$ for each $i \in \{1, 2, \ldots, n\}$. Hence, inequality (25) becomes

$$\frac{\sum_{i=1}^n \gamma_i^2}{\sum_{i=1}^n \gamma_i} - \frac{\sum_{i=1}^n \gamma_i}{n} = \frac{tr(\mathcal{C}^2)}{\varepsilon_{isi}(G)} - \frac{\varepsilon_{isi}(G)}{n} \leq (\sqrt{\gamma_1} - \sqrt{\gamma_n})^2.$$

Simplifying the above inequality, we obtain

$$(\varepsilon_{isi}(G))^2 + n(\sqrt{\gamma_1} - \sqrt{\gamma_n})^2 \varepsilon_{isi}(G) \geq ntr(\mathcal{C}^2).$$

Solving this quadratic inequality, we have

$$\varepsilon_{isi}(G) \geq \frac{1}{2}\left(\sqrt{4ntr(\mathcal{C}^2) + n^2(\sqrt{\gamma_1} - \sqrt{\gamma_n})^2} - n(\sqrt{\gamma_1} - \sqrt{\gamma_n})^2\right).$$

If $G \cong \overline{K_n}$ or $G \cong \frac{n}{2}K_2$, the equality in (26) is clearly attained.

Conversely, suppose that equality holds in (26). Then, we have that $\gamma_1 = \gamma_2 = \ldots = \gamma_n$. Therefore, $G \cong \frac{n}{2}K_2$ or $G \cong \overline{K_n}$.

This completes the proof.　□

In 1950, Biernacki, Pidek and Ryll-Nardzewski [59] proved the following Grüss-type discrete inequality.

**Lemma 21** ([59]). *Let* $x_1, x_2, \ldots, x_n$ *and* $y_1, y_2, \ldots, y_n$ *be real numbers for which there exist real constants* $a, b, A$ *and* $B$, *so that for each* $i, i = 1, 2, \ldots, n$; $a \leq x_i \leq A$ *and* $b \leq y_i \leq B$. *Then,*

$$\left|n\sum_{i=1}^n x_i y_i - \sum_{i=1}^n x_i \sum_{i=1}^n y_i\right| \leq \theta(n)(A - a)(B - b), \tag{27}$$

*where $\theta(n) = n[\frac{n}{2}](1 - \frac{1}{n}[\frac{n}{2}])$, while $[x]$ denotes the integer part of a real number $x$. Equality in Equation (27) holds if and only if $x_1 = x_2 = \ldots = x_n$ and $y_1 = y_2 = \ldots = y_n$.*

**Theorem 8.** *For any graph $G$ of order $n$, the following inequality is valid*

$$\varepsilon_{isi}(G) \geq \sqrt{ntr(\mathcal{C}^2) - \theta(n)(\gamma_1 - \gamma_n)^2}. \tag{28}$$

*The equality holds if and only if $G \cong \overline{K_n}$ or $G \cong \frac{n}{2}K_2$.*

**Proof.** Applying inequality (27) by letting $x_i = \gamma_i$, $y_i = \gamma_i$, $a = b = \gamma_n$ and $A = B = \gamma_1$, we obtain

$$\left| n \sum_{i=1}^n \gamma_i^2 - \left( \sum_{i=1}^n \gamma_i \right)^2 \right| \leq \theta(n)(\gamma_1 - \gamma_n)^2. \tag{29}$$

It implies that

$$|ntr(\mathcal{C}^2) - (\varepsilon_{isi}(G))^2| \leq \theta(n)(\gamma_1 - \gamma_n)^2.$$

From (23), we know that $\varepsilon_{isi}(G) \leq \sqrt{ntr(\mathcal{C}^2)}$. Hence, we obtain

$$\varepsilon_{isi}(G) \geq \sqrt{ntr(\mathcal{C}^2) - \theta(n)(\gamma_1 - \gamma_n)^2}.$$

Since equality in (27) holds if and only if $x_1 = x_2 = \ldots = x_n$ and $y_1 = y_2 = \ldots = y_n$, equality in (29) holds if and only if $\gamma_1 = \gamma_2 = \ldots = \gamma_n$. So, $G \cong \frac{n}{2}K_2$ or $G \cong \overline{K_n}$.
Conversely, when $G \cong \frac{n}{2}K_2$ or $G \cong \overline{K_n}$ the equality is attained.
This completes the proof. □

**Theorem 9.** *Let $G$ be a graph of order $n$ with minimum degree $\delta$. Then,*

$$\varepsilon_{isi}(G) \geq \delta \sqrt{\frac{M_1}{n}}. \tag{30}$$

*The equality holds if and only if $G \cong K_{n_1,n_2,\ldots,n_t}$, where $|n_1| = |n_2| = \ldots = |n_t|$ and $n = n_1 + n_2 + \ldots + n_t$.*

**Proof.** By Theorem 2, we have

$$\varepsilon_{isi}(G) = \sum_{i=1}^n |\mu_i| = 2 \sum_{i=1,\mu_i \geq 0}^n \mu_i \geq 2\mu_1 \geq \delta \sqrt{\frac{M_1}{n}},$$

where the second equality holds if and only if $G$ is a regular graph. Therefore, $\mu_i = \frac{\delta}{2}\lambda_i$ for any $1 \leq i \leq n$. The first equality holds if and only if $G$ has only one positive eigenvalue in its adjacency spectrum. Therefore, from Lemma 16, the equality holds if and only if $G \cong K_{n_1,n_2,\ldots,n_t}$, where $|n_1| = |n_2| = \ldots = |n_t|$.
This completes the proof. □

Given a graph $G$, if all the eigenvalues in its adjacency spectrum are nonzero, then $G$ is said to be *nonsingular*. Similarly, if all eigenvalues of the *ISI* matrix of $G$ are nonzero, then $G$ is called *ISI nonsingular*. Next, we give a lower bound on $\varepsilon_{isi}(G)$ for an *ISI* nonsingular connected graph $G$.

**Lemma 22** ([60]). *For any graph $G$ of order $n$ and size $m$, we have*

$$M_1(G) \geq \frac{4m^2}{n},$$

*and equality is attained if and only if $G$ is regular.*

**Theorem 10.** *Let G be an ISI nonsingular connected graph of order n with $\delta \geq 2$. Then, the following inequality holds*

$$\varepsilon_{isi}(G) \geq n - 1 + \frac{\delta}{2}\sqrt{\frac{M_1}{n}} + ln|det\mathcal{C}| - ln\frac{\delta}{2}\sqrt{\frac{M_1}{n}}.$$

**Proof.** Since $x \geq 1 + lnx$ for any $x > 0$, we have

$$\varepsilon_{isi}(G) = \sum_{i=1}^{n} |\mu_i| = \mu_1 + \sum_{i=2}^{n} |\mu_i|$$

$$\geq \mu_1 + \sum_{i=2}^{n} (1 + ln|\mu_i|)$$

$$= n - 1 + \mu_1 + ln \prod_{i=2}^{n} |\mu_i|$$

$$= n - 1 + \mu_1 + ln \prod_{i=1}^{n} |\mu_i| - ln\mu_1$$

$$= n - 1 + \mu_1 + ln|det\mathcal{C}| - ln\mu_1.$$

Let $f(x) = n - 1 + x + ln|det\mathcal{C}| - lnx$. It is easily seen that $f(x)$ is increasing in the variable $x \in [1, +\infty)$. By Theorem 3 and Lemma 22, we know that $\mu_1 \geq \frac{\delta}{2}\sqrt{\frac{M_1}{n}} \geq \frac{\delta m}{n} \geq 1$. Hence, we have

$$f(x) \geq f(\frac{\delta}{2}\sqrt{\frac{M_1}{n}}) = n - 1 + \frac{\delta}{2}\sqrt{\frac{M_1}{n}} + ln|det\mathcal{C}| - ln\frac{\delta}{2}\sqrt{\frac{M_1}{n}}.$$

This completes the proof. □

**Theorem 11.** *Let G be a graph of order n with minimum degree $\delta \geq 2$. Then, the following inequality is valid*

$$e^{-\sqrt{tr(\mathcal{C}^2)}} \leq \varepsilon_{isi}(G) \leq e^{\sqrt{tr(\mathcal{C}^2)}}.$$

**Proof.** Since $x < e^x$ for any real number $x$, it follows that

$$\varepsilon_{isi}(G) = \sum_{i=1}^{n} |\mu_i| < \sum_{i=1}^{n} e^{|\mu_i|} = \sum_{i=1}^{n} \sum_{k \geq 0} \frac{(|\mu_i|)^k}{k!} = \sum_{k \geq 0} \frac{1}{k!} \sum_{i=1}^{n} (|\mu_i|)^k.$$

From Lemma 15, we have

$$\varepsilon_{isi}(G) < \sum_{k \geq 0} \frac{1}{k!} \sum_{i=1}^{n} (|\mu_i|)^k$$

$$\leq \sum_{k \geq 0} \frac{1}{k!} \left( \sum_{i=1}^{n} (|\mu_i|^2) \right)^{\frac{k}{2}}$$

$$= \sum_{k \geq 0} \frac{1}{k!} \left( \sqrt{tr(\mathcal{C}^2)} \right)^k = e^{\sqrt{tr(\mathcal{C}^2)}}.$$

Let $l$ be the number of nonzero eigenvalues of the matrix $\mathcal{C}$, and let $\theta_1, \theta_2, \ldots, \theta_l$ be the absolute values of all these nonzero eigenvalues, given in a non-increasing order. By Lemmas 1 and 7,

$$\mu_n \leq \rho_2[\mathcal{C}_2] \leq -\frac{d_i d_j}{d_i + d_j} \leq -\frac{\delta}{2},$$

where $[\mathcal{C}_2]$ is the leading $2 \times 2$ submatrix of $\mathcal{C}$. Therefore, $|\mu_n| \geq 1$. Hence,

$$\sum_{i=1}^{l} \theta_i = \sum_{i=1}^{n} |\mu_i| \geq |\mu_n| \geq 1 \,. \tag{31}$$

Using the arithmetic–geometric mean inequality, we have

$$\varepsilon_{isi}(G) = \sum_{i=1}^{n} |\mu_i| = \sum_{i=1}^{l} \theta_i = l \left( \sum_{i=1}^{l} \frac{1}{l} \theta_i \right) \geq l \left( \sqrt[l]{\theta_1 \theta_2 \dots \theta_l} \right).$$

It follows from Lemmas 3 and 17 and Equation (30) that

$$l \left( \sqrt[l]{\theta_1 \theta_2 \dots \theta_l} \right) \geq l \left( \frac{l}{\sum_{i=1}^{l} \frac{1}{\theta_i}} \right)$$

$$\geq l \left( \frac{l}{\sum_{i=1}^{l} \frac{1}{\theta_i} \sum_{i=1}^{l} \theta_i} \right)$$

$$\geq l \left( \frac{l}{l \sum_{i=1}^{l} \frac{1}{\theta_i} \theta_i} \right).$$

Applying the power series expansion of $e^x$, we obtain

$$l \left( \frac{l}{l \sum_{i=1}^{l} \frac{1}{\theta_i} \theta_i} \right) \geq l \left( \frac{l}{l^2 \sum_{i=1}^{l} \theta_j} \right) > \frac{1}{\sum_{i=1}^{l} e^{\theta_i}}$$

$$= \frac{1}{\sum_{i=1}^{l} \sum_{k \geq 0} \frac{(\theta_i)^k}{k!}} = \frac{1}{\sum_{k \geq 0} \frac{1}{k!} \sum_{i=1}^{l} (\theta_i)^k}.$$

It follows from Lemma 15 that

$$\frac{1}{\sum_{k \geq 0} \frac{1}{k!} \sum_{i=1}^{l} (\theta_i)^k} \geq \frac{1}{\sum_{k \geq 0} \frac{1}{k!} \sum_{i=1}^{l} ((\theta_i)^2)^{\frac{k}{2}}}$$

$$= \frac{1}{\sum_{k \geq 0} \frac{1}{k!} (\sqrt{tr(\mathcal{C}^2)})^k} = e^{-\sqrt{tr(\mathcal{C}^2)}} \,.$$

This completes the proof. $\square$

**Theorem 12.** *Let G be a connected graph of order $n > 1$ with m edges and minimum degree $\delta$. Then,*

$$\varepsilon_{isi}(G) \geq \delta \sqrt{m} \,, \tag{32}$$

*and the equality holds if and only if $G \cong K_{\frac{n}{2}, \frac{n}{2}}$.*

**Proof.** For $n = 2$, $G = K_{1,1}$ and, hence, the equality holds. Otherwise, $n \geq 3$. From Lemma 14, we know that the sum of the eigenvalues of $\mathcal{C}$ is zero, and we can deduce that

$$0 = \left( \sum_{i=1}^{n} \mu_i \right)^2 = \sum_{i=1}^{n} \mu_i^2 + 2 \sum_{1 \leq i < j \leq n} \mu_i \mu_j.$$

Therefore, we have

$$\sum_{i=1}^{n} \mu_i^2 = 2 \left| \sum_{1 \leq i < j \leq n} \mu_i \cdot \mu_j \right|.$$

Combining the definition of ISI energy and Lemmas 1 and 14, we obtain

$$(\varepsilon_{isi}(G))^2 = \left( \sum_{i=1}^{n} |\mu_i| \right)^2$$

$$= \sum_{i=1}^{n} |\mu_i|^2 + 2 \sum_{1 \le i < j \le n} |\mu_i| \cdot |\mu_j|$$

$$\ge \sum_{i=1}^{n} |\mu_i|^2 + 2 \left| \sum_{1 \le i < j \le n} \mu_i \cdot \mu_j \right| \tag{33}$$

$$= 2 \sum_{i=1}^{n} \mu_i^2 = 4 \sum_{v_i v_j \in E(G)} \frac{d_i^2 d_j^2}{(d_i + d_j)^2} \ge m\delta^2, \tag{34}$$

and inequality (32) follows. This concludes the first part of the proof.

Suppose now that the equality holds in (32). Then, all the above inequalities must be equalities. Equality in (34) implies that $\frac{d_i^2 d_j^2}{(d_i+d_j)^2} = \frac{\delta^2}{4}$ for each edge $v_i v_j \in E(G)$; that is, $d_i = d_j$ for each edge $v_i v_j \in E(G)$. As $G$ is assumed to be connected, it is regular.

From equality in (33), we see that there are two nonzero eigenvalues and all the remaining eigenvalues are zero; that is, $\mu_1 = -\mu_n$ and $\mu_i = 0$ for $2 \le i \le n-1$. Since $G$ is regular, $\frac{\delta}{2}\lambda_i = \mu_i$ for all $1 \le i \le n$. Therefore, $\lambda_1 = -\lambda_n$ and $\lambda_i = 0$ for $2 \le i \le n-1$. Since $G$ is connected, by Lemma 18, it must be $G \cong K_{\frac{n}{2},\frac{n}{2}}$.

Conversely, by direct checking, we verify that equality holds in (31) for $G \cong K_{\frac{n}{2},\frac{n}{2}}$.

This completes the proof.　□

**Theorem 13.** *Let $G$ be a graph of order $n > 1$ and size $m$. Then,*

$$\varepsilon_{isi}(G) \ge \delta(G')\sqrt{m}, \tag{35}$$

*where $G'$ is the graph obtained from $G$ by deleting all isolated vertices. The equality holds if and only if $G \cong \overline{K_n}$ or $G \cong K_{p,p} \cup (n - 2p)K_1$ and $p = 1, 2, \ldots, \lfloor \frac{n}{2} \rfloor$.*

**Proof.** For $m = 0$, we have $G = \overline{K_n}$ and, hence, the equality holds. Otherwise, $m \ge 1$. Let $p$ be the number of isolated vertices and let $k$ be the number of connected components in $G$. In addition, let $G_i$ be the $i$-th connected component of $G$ with order $n_i \ge 2$, $m_i \ge 1$ edges and minimum degree $\delta_i$. Hence, we have $n = p + \sum_{i=1}^{k} n_i$, $m = \sum_{i=1}^{k} m_i$ and $\delta_i \ge \delta(G')$. Without loss of generality, we may assume that $m_1 \ge m_2 \ge \ldots \ge m_k \ge 1$. By Theorem 12, we have

$$\varepsilon_{isi}(G_i) \ge \delta_i \sqrt{m_i} \ge \delta(G')\sqrt{m_i}. \tag{36}$$

Notice that, for positive real numbers $a$ and $b$, $(a \ge b)$,

$$\sqrt{a} + \sqrt{b} \ge \sqrt{a + b}, \tag{37}$$

with equality if and only if $b = 0$. Applying this result to (36) and by Lemma 19, we obtain

$$\varepsilon_{isi}(G) = \sum_{i=1}^{k} \varepsilon_{isi}(G_i)$$

$$\ge \delta(G')\sqrt{m_1} + \delta(G')\sqrt{m_2} + \ldots + \delta(G')\sqrt{m_k}$$

$$\ge \delta(G')\sqrt{m_1 + m_2} + \delta(G')\sqrt{m_3} + \ldots + \delta(G')\sqrt{m_k}$$

$$\ge \delta(G')\sqrt{m_1 + m_2 + m_3} + \delta(G')\sqrt{m_4} + \ldots + \delta(G')\sqrt{m_k}$$

$$\ldots \ldots$$

$$\geq \delta(G')\sqrt{m_1 + m_2 + \ldots + m_k} = \delta(G')\sqrt{m} \, .$$

This concludes the first part of the proof.

Suppose that the equality holds in (35) for $m \geq 1$. Then, all the above inequalities must be equalities. Since $m_i \geq 1$, $1 \leq i \leq k$, we must have $k = 1$. By Theorem 12, we then have $G_1 \cong K_{\frac{n_1}{2}, \frac{n_1}{2}}$. Hence, $G \cong K_{p,p} \cup (n - 2p)K_1$ for $n_1 = p = 1, 2, \ldots, \lfloor \frac{n}{2} \rfloor$.

Conversely, one can easily see that the equality holds in (35) for $G \cong K_{p,p} \cup (n - 2p)K_1$, $p = 1, 2, \ldots, \lfloor \frac{n}{2} \rfloor$.

This completes the proof. $\square$

**Theorem 14.** *Let $G$ be a connected graph of order $n > 1$ with $m$ edges and minimum degree $\delta$. Then,*

$$\varepsilon_{isi}(G) \geq |\mu_n| + \sqrt{m\delta^2 - 3\mu_n^2} \, , \tag{38}$$

*and the equality holds if and only if $G \cong K_{\frac{n}{2}, \frac{n}{2}}$.*

**Proof.** From Lemma 14, we know that the sum of the eigenvalues of $\mathcal{C}$ is zero. We can deduce

$$\left( \sum_{i=1}^{n-1} \mu_i \right)^2 = \sum_{i=1}^{n-1} \mu_i^2 + 2 \sum_{1 \leq i < j \leq n} \mu_i \cdot \mu_j = \mu_n^2$$

and

$$\left( \sum_{i=1}^{n-1} |\mu_i| \right)^2 = \sum_{i=1}^{n-1} |\mu_i|^2 + 2 \sum_{1 \leq i < j \leq n} |\mu_i| \cdot |\mu_j|.$$

Bearing these identities in mind, we obtain

$$\left( \varepsilon_{isi}(G) - |\mu_n| \right)^2 = \sum_{i=1}^{n-1} |\mu_i|^2 + 2 \sum_{1 \leq i < j \leq n} |\mu_i| \cdot |\mu_j|$$

$$\geq \sum_{i=1}^{n-1} |\mu_i|^2 + 2 \left| \sum_{1 \leq i < j \leq n} \mu_i \cdot \mu_j \right| \tag{39}$$

$$= \sum_{i=1}^{n-1} |\mu_i|^2 + \left| \mu_n^2 - \sum_{i=1}^{n-1} \mu_i^2 \right|.$$

One can easily see that $\mu_n^2 \leq \frac{1}{2} \sum_{i=1}^{n} |\mu_i|^2$. In view of this, we have

$$\left( \varepsilon_{isi}(G) - |\mu_n| \right)^2 \geq 2 \sum_{i=1}^{n} \mu_i^2 - 3\mu_n^2 \, .$$

Therefore, we have

$$\varepsilon_{isi}(G) \geq |\mu_n| + \sqrt{2tr(\mathcal{C}^2) - 3\mu_n^2} \, .$$

Lemmas 1 and 14 imply that

$$tr(\mathcal{C}^2) = 2 \sum_{v_i v_j \in E(G)} \frac{d_i^2 d_j^2}{(d_i + d_j)^2} \geq \frac{m\delta^2}{2}. \tag{40}$$

Hence, we have

$$\varepsilon_{isi}(G) \geq |\mu_n| + \sqrt{m\delta^2 - 3\mu_n^2} \, .$$

Suppose now that the equality holds in (38). Then, all the above inequalities must be equalities. Equality in (40) implies that $\frac{d_i^2 d_j^2}{(d_i+d_j)^2} = \frac{\delta^2}{4}$ for each edge $v_i v_j \in E(G)$; that is, $d_i = d_j$, for each edge $v_i v_j \in E(G)$. As $G$ is assumed to be connected, it is regular.

From equality in (39), we see that there are two nonzero eigenvalues and all the remaining eigenvalues are zero; that is, $\mu_1 = -\mu_n$ and $\mu_i = 0$ for $2 \leq i \leq n-1$. Since $G$ is regular, $\frac{\delta}{2}\lambda_i = \mu_i$ for all $1 \leq i \leq n$. Therefore, $\lambda_1 = -\lambda_n$ and $\lambda_i = 0$ for $2 \leq i \leq n-1$. Since $G$ is connected, by Lemma 18, we conclude that $G \cong K_{\frac{n}{2},\frac{n}{2}}$.

Conversely, by direct checking we verify that equality holds in (38) for $G \cong K_{\frac{n}{2},\frac{n}{2}}$.

This completes the proof. $\square$

Before proving the next theorems, we need the following lemma.

**Lemma 23.** *Let $G$ be a graph with $n$ vertices and let $-b_1 \leq \ldots \leq -b_{n_2} \leq \ldots \leq a_{n_1} \leq \ldots \leq a_1$ be the eigenvalues of the ISI matrix $\mathcal{C}$ of $G$, where $a_{n_1}$ is non-negative and $b_{n_2}$ is positive. Then,*

$$\varepsilon_{isi}(G) = \sqrt{2tr(\mathcal{C}^2) + 4\left(\sum_{1 \leq i_1 < i_2 \leq n_1} a_{i_1} a_{i_2} + \sum_{1 \leq j_1 < j_2 \leq n_2} b_{j_1} b_{j_2}\right)}.$$

**Proof.** From Lemma 14, we know that the sum of the eigenvalues of $\mathcal{C}$ is zero, so we can deduce

$$\sum_{i=1}^{n_1} a_i = \sum_{j=1}^{n_2} b_j.$$

Then,

$$(\varepsilon_{isi}(G))^2 = \left(\sum_i a_i + \sum_j b_j\right)^2 = 2\left((\sum_i a_i)^2 + (\sum_j b_j)^2\right)$$

$$= 2\left(\sum_i a_i^2 + \sum_j b_j^2 + 2\sum_{1 \leq i_1 < i_2 \leq n_1} a_{i_1} a_{i_2} + 2\sum_{1 \leq j_1 < j_2 \leq n_2} b_{j_1} b_{j_2}\right)$$

$$= 2tr(\mathcal{C}^2) + 4\left(\sum_{1 \leq i_1 < i_2 \leq n_1} a_{i_1} a_{i_2} + \sum_{1 \leq j_1 < j_2 \leq n_2} b_{j_1} b_{j_2}\right).$$

This completes the proof. $\square$

**Theorem 15.** *Let $G$ be a graph of order $n$, and let the absolute values of the eigenvalues of the ISI matrix $\mathcal{C}$ of $G$ be $\gamma_1 \geq \gamma_2 \geq \ldots \geq \gamma_n$. Then, the following inequality is valid:*

$$\varepsilon_{isi}(G) \geq \frac{1}{2}\left(\gamma_n(n-2) + \sqrt{8tr(\mathcal{C}^2) + (\gamma_n)^2(n-2)^2}\right). \tag{41}$$

*The equality holds if and only if $G \cong \overline{K_n}$ or $G \cong \frac{n}{2}K_2$.*

**Proof.** We use the notations of Lemma 23. Let $-b_1 \leq \ldots \leq -b_{n_2} \leq \ldots \leq a_{n_1} \leq \ldots \leq a_1$ be the eigenvalues of the ISI matrix $\mathcal{C}$ of $G$, where $a_{n_1}$ is non-negative and $b_{n_2}$ is positive. Then, $\gamma_n = \min\{a_1, \ldots, a_{n_1}, b_1, \ldots, b_{n_2}\}$.

It is obvious that $a_{i_1}, a_{i_2} \geq \gamma_n$. Therefore, we have

$$\left(a_{i_1} - \frac{\gamma_n}{2}\right)\left(a_{i_2} - \frac{\gamma_n}{2}\right) \geq \frac{\gamma_n}{4},$$

i.e.,

$$a_{i_1} a_{i_2} \geq \frac{\gamma_n}{2}(a_{i_1} + a_{i_2}). \tag{42}$$

By similar arguments, we can obtain

$$b_{j_1} b_{j_2} \geq \frac{\gamma_n}{2} (b_{j_1} + b_{j_2}) .$$ (43)

Combining Lemma 23 with the fact that $\sum_i a_i = \sum_j b_j = \frac{\varepsilon_{isi}(G)}{2}$, we can deduce

$$(\varepsilon_{isi}(G))^2 \geq 2tr(\mathcal{C}^2) + 2\gamma_n \left( \sum_{1 \leq i_1 < i_2 \leq n_1} (a_{i_1} + a_{i_2}) + \sum_{1 \leq j_1 < j_2 \leq n_2} (b_{j_1} + b_{j_2}) \right)$$

$$= 2tr(\mathcal{C}^2) + 2\gamma_n \left( (n_1 - 1) \sum_i a_i + (n_2 - 1) \sum_j b_j \right)$$

$$= 2tr(\mathcal{C}^2) + (n-2)\gamma_n \varepsilon_{isi}(G).$$

By solving this quadratic inequality, we obtain the result

$$\varepsilon_{isi}(G) \geq \frac{1}{2} \left( \gamma_n(n-2) + \sqrt{8tr(\mathcal{C}^2) + (\gamma_n)^2(n-2)^2} \right).$$

Suppose that the equality holds in (41). Then, all the above inequalities (42) and (43) must be equalities, and we have $a_1 = \ldots = a_{n_1} = -b_1 = \ldots = -b_{n_2}$, i.e., $\gamma_1 = \gamma_2 = \ldots = \gamma_n$. Thus, by Lemma 13, $G \cong \frac{n}{2} K_2$ or $G \cong \overline{K_n}$.

Conversely, one can easily see that the equality holds in (41) for $G \cong \frac{n}{2} K_2$ or $G \cong \overline{K_n}$. This completes the proof. $\square$

Consider a graph whose eigenvalues are not in the interval $(-1, 1)$. In the next theorem, we give a lower bound for the energy of such a graph.

**Theorem 16.** *Let $G$ be a graph of order $n$ with $n_1$ non-negative eigenvalues such that $\gamma_1 \geq \gamma_2 \geq \ldots \geq \gamma_n \geq 1$. Then*

$$\varepsilon_{isi}(G) \geq \sqrt{2tr(\mathcal{C}^2) + 4\gamma_1(n_1 - 1) + (n-1)(n-3)(\gamma_n)^2} .$$

**Proof.** We use the notations of Lemma 23. Let $-b_1 \leq \ldots \leq -b_{n_2} \leq \ldots \leq a_{n_1} \leq \ldots \leq a_1$ be the eigenvalues of the ISI matrix $\mathcal{C}$ of $G$, where $a_{n_1}$ is non-negative and $b_{n_2}$ is positive. Then, $\gamma_1 = max\{a_1, \ldots, a_{n_1}, b_1, \ldots, b_{n_2}\}$ and $\gamma_n = min\{a_1, \ldots, a_{n_1}, b_1, \ldots, b_{n_2}\} \geq 1$. Since $G$ has no eigenvalue in the interval $[0, 1)$, then

$$\sum_{i_1 < i_2} a_{i_1} a_{i_2} \geq \gamma_1 \sum_{i_2 < n_1 - 1} a_{i_2} + \binom{n_1 - 1}{2} (\gamma_n)^2$$

$$\geq \gamma_1(n_1 - 1) + \binom{n_1 - 1}{2} (\gamma_n)^2,$$

and

$$\sum_{j_1 < j_2} b_{j_1} b_{j_2} \geq \binom{n_2}{2} (\gamma_n)^2.$$

By Lemma 23, we know that

$$(\varepsilon_{isi}(G))^2 = 2tr(\mathcal{C}^2) + 4 \left( \sum_{i_1 < i_2} a_{i_1} a_{i_2} + \sum_{j_1 < j_2} b_{j_1} b_{j_2} \right)$$

$$\geq 2tr(\mathcal{C}^2) + 4\gamma_1(n_1 - 1) + 4(\gamma_n)^2 \left( \binom{n_1 - 1}{2} + \binom{n_2}{2} \right).$$

It is easy to prove that

$$\binom{n_1 - 1}{2} + \binom{n_2}{2} = \frac{(n_1 - 1)(n_1 - 2)}{2} + \frac{n_2(n_2 - 1)}{2}$$

$$= \frac{(n_1 - 1)^2 + (n_2)^2}{2} - \frac{n_1 + n_2 - 1}{2}$$

$$\geq \frac{(n_1 + n_2 - 1)^2}{4} - \frac{n - 1}{2} = \frac{(n - 1)^2}{4} - \frac{n - 1}{2}$$

$$= \frac{(n - 1)(n - 3)}{4}.$$

Thus, we have

$$(\varepsilon_{isi}(G))^2 \geq 2tr(\mathcal{C}^2) + 4\gamma_1(n_1 - 1) + (n - 1)(n - 3)(\gamma_n)^2.$$

This completes the proof. $\square$

**Lemma 24** ([61]). *Let G be a graph where the number of eigenvalues greater than, less than, and equal to zero are p, q and r, respectively. Then,*

$$\alpha \leq r + min\{p, q\},$$

*where α is the independence number of G.*

**Theorem 17.** *Let G be a graph of order n, where the number of eigenvalues of the ISI matrix C greater than, less than, and equal to zero are $n_1$, $n_2$ and r, respectively. Let α denote the independence number of G. Then, the following inequality is valid:*

$$\varepsilon_{isi}(G) \leq \sqrt{2(n - \alpha)tr(\mathcal{C}^2)} . \tag{44}$$

*The equality holds if and only if $G \cong \overline{K_n}$ or $G \cong \frac{n}{2}K_2$ .*

**Proof.** Let $a_{n_1} \leq \ldots \leq a_1$ be the $n_1$ positive eigenvalues, and let $-b_1 \leq \ldots \leq -b_{n_2}$ be the $n_2$ negative eigenvalues of the ISI matrix $\mathcal{C}$ of $G$. Then, $\mathcal{C}$ has $r = n - n_1 - n_2$ eigenvalues which are equal to zero. By Lemma 24, we know that

$$\alpha \leq (n - n_1 - n_2) + min\{n_1, n_2\}.$$

Therefore, $\alpha \leq (n - n_1 - n_2) + n_1$ and $\alpha \leq (n - n_1 - n_2) + n_2$, i.e., $n_1 \leq n - \alpha$ and $n_2 \leq n - \alpha$. Since

$$\sum_{i=1}^{n_1} a_i - \sum_{j=1}^{n_2} b_j = 0 ,$$

we have

$$\varepsilon_{isi}(G) = 2\sum_{i=1}^{n_1} a_i = 2\sum_{j=1}^{n_2} b_j.$$

Furthermore, by Lemma 2, we obtain

$$\varepsilon_{isi}(G) = 2\sum_{i=1}^{n_1} a_i \leq 2\sqrt{n_1 \sum_{i=1}^{n_1} a_i^2} , \tag{45}$$

and

$$\varepsilon_{isi}(G) = 2\sum_{j=1}^{n_2} b_j \leq 2\sqrt{n_2 \sum_{j=1}^{n_2} b_j^2} \ . \tag{46}$$

Therefore,

$$\frac{(\varepsilon_{isi}(G))^2}{2} = \frac{(\varepsilon_{isi}(G))^2}{4} + \frac{(\varepsilon_{isi}(G))^2}{4}$$

$$\leq n_1 \sum_{i=1}^{n_1} a_i^2 + n_2 \sum_{j=1}^{n_2} b_j^2$$

$$\leq (n-\alpha) \sum_{i=1}^{n_1} a_i^2 + (n-\alpha) \sum_{j=1}^{n_2} b_j^2$$

$$= (n-\alpha)\left(\sum_{i=1}^{n_1} a_i^2 + \sum_{j=1}^{n_2} b_j^2\right)$$

$$= (n-\alpha)tr(\mathcal{C}^2) \ .$$

Hence, we have

$$\varepsilon_{isi}(G) \leq \sqrt{2(n-\alpha)tr(\mathcal{C}^2)} \ .$$

If the equality holds, then equalities in both (45) and (46) hold. Therefore, we have $a_1 = \ldots = a_{n_1} = b_1 = \ldots = b_{n_2}$, Hence, $G \cong \frac{n}{2}K_2$ or $G \cong \overline{K_n}$.

Conversely, when $G \cong \frac{n}{2}K_2$ or $G \cong \overline{K_n}$ the equality is attained.

This completes the proof. □

## 5. Conclusions

In theoretical chemistry, topological indices are utilized for indicating the physical and chemical properties of molecules. Among the considerable number of topological indices, the ISI index has a great advantage in forecasting the overall superficial area of octane isomers. Graph energy, a parameter found to be closely interrelated with topological indices, has been comprehensively and deeply investigated, on account of the fact that it approximates to the total $\pi$-electron energy of a molecule. The utilization of graph energies is not only in chemistry, but also in unforeseen fields, including air transportation, satellite communication, face recognition, crystallography, etc. It is noted that energy of many kinds of graphs can be determined by their ISI energy $\varepsilon_{isi}$. Hence, we consider the $\varepsilon_{isi}$ of graphs and establish several new sharp bounds for $\varepsilon_{isi}$ and $\mu_1$ in the light of $\mathcal{C}(G)$, $M_1(G)$ and $M_2(G)$, $\alpha(G)$, and other graph parameters, and we give descriptions of the corresponding extremal graphs.

Trees, chemical trees, and unicyclic and bicyclic graphs are common models of chemical structures. Therefore, studying the $\varepsilon_{isi}$ of these graphs is interesting in future.

Let $G_1$ and $G_2$ be two $n$-vertex nonisomorphic graphs, we call $G_1$ and $G_2$ ISI-cospectral if $Sp_{isi}(G_1) = Sp_{isi}(G_2)$. $G_1$ and $G_2$ are said to be ISI-equienergetic if $\varepsilon_{isi}(G_1) = \varepsilon_{isi}(G_2)$. Hence, constructing ISI-noncospectral and ISI-equienergetic chemical trees, line graphs and other useful graphs is also an interesting research direction.

**Author Contributions:** F.L., Q.Y. and H.B. contributed equally to conceptualization, methodology, validation, formal analysis, writing-review and editing; project administration, F.L. All authors read and approved the final manuscript.

**Funding:** This research received no external funding.

**Institutional Review Board Statement:** Not applicable.

**Informed Consent Statement:** Not applicable.

**Data Availability Statement:** Not applicable.

**Acknowledgments:** The authors are very grateful to anonymous referees and editors for their constructive suggestions and insightful comments, which have considerably improved the presentation of this paper.

**Conflicts of Interest:** The authors declare no conflict of interest.

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
