# Peer review of "Some New Bounds for the Inverse Sum Indeg Energy of Graphs"

_axioms, doi:10.3390/axioms11050243_

Round 1

Reviewer 1 Report

No comment

Author Response

Thank you very much for your kindness and we would like to
thank you for your useful comments and valuable suggestions. I
have carefully read your reports and revised the manuscript
according to your suggestions.

Reviewer 2 Report

In the manuscript, the authors study the Inverse sum indeg energy of graphs. In particular, they give some upper and lower bounds for the spectral radius of the ISI matrix and characterizes the extremal graphs. In addition, they obtain some bounds on the ISI energy.

The manuscript is well written, the results seem to be correct and provide an acceptable contribution. However, I consider that the importance of the paper should be highlighted in the conclusions. Perhaps some open problems can be added, or some new research line to continue deepening.
Hence, I recommend the publication of this manuscript, after a minor revision of the conclusions.

Author Response

Dear reviewers,

Thank you very much for your kindness and we would like to
thank you for your useful comments and valuable suggestions. I
have carefully read your reports and revised the manuscript
according to your suggestions.

All problems mentioned in the reports are answered as follows.

1. Problem: 2/27\ldots Please give a space behind commas: [1,9,10 ...] ...please correct into: [1, 9, 10, 17,  ...   ].

Answer: I try to correct [1,9,10 ...] into: [1, 9, 10, 17,  ...   ]. But the Latex can not support to give space behind commas in [1,9,10 ...], so, I correct it into [1], [9], [10], [17], ... .

2. 5/74...equation (13) Please give a dot at the end of the formula but correct way.

Answer: I have put a dot at the end of the formula in correct way. And I have corrected all other similar errors throughout this paper.

3. 12/144 Please check the end of the formula...and give more space before a dot, which is ending formula.

Answer: I have corrected all such similar errors throughout this paper.

4. Introduction....Please correct introduction...extend it and present more deeply history
and motivation. Discuss as well new results of Kinkar Das, Furtula, Vetrik Tomas, Mesfin
Masre, Elias John and others. They all have connections to graph indices so make introduction
more relevant. longer  and deeply. .

Answer: I have added some more new results of Kinkar Das, Furtula, Vetrik Tomas, Mesfin Masre, Elias John and others, and give more history and motivation of energy and ISI energy research.

5. In Conclusion: Please insert some new ideas and open questions, lines devoted to
this line of research - the first and second Zagreb indices of graphs.
Please include a discussion and real applications of ISI energy of graphs.

Answer: I have rewrite the Conclusion according to your good suggestions.

6. However, the Conclusions chapter is too short and reflects the authors' opinion on the field, applications in the real world of research, future research. This chapter can be improved.

Answer: I have rewrite the Conclusion according to your good suggestions.

7. However, I consider that the importance of the paper should be highlighted in the conclusions. Perhaps some open problems can be added, or some new research line to continue deepening..

Answer: I have rewrite the Conclusion according to your good suggestions.

8. I added ``The authors are very grateful to anonymous referees and editor for their
constructive suggestions and insightful comments comments, which have considerably improved the
presentation of this paper." to the Acknowledgments.

9. I added some new references about energy and topological indices of graphs.

Please contact me if any problems.
Kind regards!
Fengwei Li
College of Basic Science
Ningbo University of Finance \& Economics,
Ningbo, Zhejiang 315175, P.R. China.
fengwei.li@hotmail.com

Reviewer 3 Report

The research is interesting. The authors use the concept of energy of the Graph. The demonstrations are made for every Theorems and Lemmas. The Introductions chapter is well done and introduces the reader to the context of the research. The literature review covers the study area. The applications in Chapter 4 are well ranked and the authors prove the Theorems and Lemmas expressed.
However, the Conclusions chapter is too short and reflects the authors' opinion on the field, applications in the real world of research, future research. This chapter can be improved

Author Response

(The authors gave the same response as above.)

Reviewer 4 Report

The authors consider and study several new bounds for the Inverse Sum Indeg Energy of Graphs. They bring new results and the research topic is chosen well-it is very hot topic. According my opinion the paper is good, but it needs some improvements. I recommend the publication of the paper under major revisions listed bellow:

page/line

2/27.....Please give  a space behind commas:  [1,9,10 ...] ...please correct into: [1, 9, 10, 17,     ...    ]

5/74...equation (13) Please give a dot at the end of the formula but correct way. 

12/144 Please check the end of the formula...and give more space before  a dot, which is ending formula.

Introduction....Please correct introduction...extend it and present more deeply history and motivation. Discuss as well new results of Kinkar Das, Furtula, Vetrik Tomas, Mesfin Masre, Elias John and others. They all have connections to graph indices so make introduction more relevant. longer  and deeply. 

Conclusion:

Please insert some new ideas and open questions, lines devoted to this line of research - the first and second Zagreb indices of graphs.

Please include a discussion and real applications of ISI energy of graphs. 

Author Response

(The authors gave the same response as above.)

Round 2

Reviewer 4 Report

I agree with corrected version. I recommend for publication in the present form.